# Differences in the Emission of Volatile Organic Compounds (VOCs) between Non-Differentiating and Adipogenically Differentiating Mesenchymal Stromal/Stem Cells from Human Adipose Tissue

**DOI:** 10.3390/cells8070697

**Published:** 2019-07-10

**Authors:** Ann-Christin Klemenz, Juliane Meyer, Katharina Ekat, Julia Bartels, Selina Traxler, Jochen K. Schubert, Günter Kamp, Wolfram Miekisch, Kirsten Peters

**Affiliations:** 1Department of Anesthesiology and Intensive Care Medicine, University Medical Centre Rostock, Schillingallee 35, 18057 Rostock, Germany; 2Department of Cell Biology, University Medical Centre Rostock, Schillingallee 69, 18057 Rostock, Germany; 3AMP-Lab GmbH, Mendelstr. 11, 48149 Münster, Germany

**Keywords:** adipose tissue-derived mesenchymal stromal/stem cells (ASCs), cell differentiation, volatile organic compounds, metabolic monitoring

## Abstract

Metabolic characterization of human adipose tissue-derived mesenchymal stromal/stem cells (ASCs) is of importance in stem cell research. The monitoring of the cell status often requires cell destruction. An analysis of volatile organic compounds (VOCs) in the headspace above cell cultures might be a noninvasive and nondestructive alternative to in vitro analysis. Furthermore, VOC analyses permit new insight into cellular metabolism due to their view on volatile compounds. Therefore, the aim of our study was to compare VOC profiles in the headspace above nondifferentiating and adipogenically differentiating ASCs. To this end, ASCs were cultivated under nondifferentiating and adipogenically differentiating conditions for up to 21 days. At different time points the headspace samples were preconcentrated by needle trap micro extraction and analyzed by gas chromatography/mass spectrometry. Adipogenic differentiation was assessed at equivalent time points. Altogether the emissions of 11 VOCs showed relevant changes and were analyzed in more detail. A few of these VOCs, among them acetaldehyde, were significantly different in the headspace of adipogenically differentiating ASCs and appeared to be linked to metabolic processes. Furthermore, our data indicate that VOC headspace analysis might be a suitable, noninvasive tool for the metabolic monitoring of (mesenchymal stem) cells in vitro.

## 1. Introduction

Due to their capacity for self-renewal and multipotent differentiation, mesenchymal stem/stromal cells (MSCs) have been identified as playing an essential role in tissue homeostasis and regeneration [1]. In recent years, increasing attention has been paid to MSCs from human adipose tissue (adipose tissue-derived MSCs, referred to as ASCs), as they show promising potential as a clinical alternative to other MSCs, like bone marrow-derived MSCs (bmMSCs) [2,3,4]. Zuk et al. were the first to characterize the mesenchymal differentiation potential of ASCs in more detail [5,6]. Furthermore, nonmesenchymal differentiation (e.g., neuron-like morphology) of ASCs has also been demonstrated [7].

Data concerning the energy metabolism of MSCs in vitro are available but have not been worked out in detail. It has been shown that bmMSCs mainly facilitate glycolysis with subsequent lactate production rather than oxidative phosphorylation [8]. Furthermore, it was demonstrated that nondifferentiating and osteogenically differentiating bmMSCs facilitated oxidative phosphorylation as well as glycolysis in order to fulfill their energy metabolic needs [9]. It has been reported that the function of mitochondria also regulates the differentiation of MSC [10,11]. Adipogenic differentiation of ASC is accompanied by increasing mitochondrial enzyme activities, indicating a growing capacity for oxidative phosphorylation and β-oxidation and, thus, a shift towards lipid metabolism [12]. In addition, in human differentiated adipocytes, the enzymes involved in mitochondrial metabolism showed significantly higher activity rates than in the CD34^+^ stromal vascular fraction of human adipose tissue containing the nondifferentiated progenitors of adipocytes, the ASCs [13]. Thus, in recent years the idea matured that metabolic pathways regulate cellular differentiation processes that go beyond ATP production [14]. To determine metabolic activities in different phases of differentiation, further information is needed. Ideally, gathering this data should not influence or even disturb the cell culture.

For conventional metabolomics studies in vitro, sufficient cell material is necessary. Especially when working with human stem cells, often low cell numbers are available. In contrast to those conventional methods, an analysis of volatile organic compounds (VOCs) has the great advantage of being nondestructive and noninvasive [15]. Such a destruction-free analysis can be done by means of bidirectional preconcentration combined with a versatile, standardized set-up and has, therefore, gained more attention during the last decades. Besides the occurrence of VOCs in human breath, they are also emitted in trace concentrations by cell and bacteria cultures [16,17,18]. 

Headspace analysis by means of needle trap microextraction (NTME) coupled with gas chromatography and mass spectrometry (e.g., GC-MS with electron impact quadrupole detection) allows the identification and quantification of compounds down to the parts per trillion by volume (pptV) range. Conventional headspace techniques require larger volumes and, therefore, may affect the investigated system itself. Hence, microextraction techniques are better suited for this kind of setup [19]. In contrast to solid phase microextraction (SPME), the sensitivity of NTME can be enhanced by increasing the sampling volume, as the sample volume is usually limited. Bidirectional NTME sampling can be applied to reach a high sensitivity in the VOC extraction [20] without significant effects on volume or pressure of the in vitro system. A crucial step for a reliable and reproducible VOC headspace analysis in in vitro cultures is also a standardized set-up [21]. 

Data from several studies suggest that different cell lines and bacterial strains can be distinguished from each other by means of their volatile emissions [22,23,24]. Some volatiles have been shown to be associated with different pathways in cell metabolism [25]. For example, acetone is produced by the decarboxylation of acetoacetate, and acetaldehyde is produced by ethanol oxidation [26]. Previous work has established that volatile substances over cell cultures correlate with stem cell growth [27]. 

Therefore, the aim of our study was to assess VOC emissions in the headspace of cell cultures to gather complementary information on metabolic changes of ASCs during differentiation. This raises the question of whether we can detect the differences in VOC emissions from cell cultures between nondifferentiating and differentiating ASCs. This in turn leads to the next question: can the differences in VOC emission be related to metabolic changes in ASCs during differentiation? 

## 2. Materials and Methods

### 2.1. Isolation, Cultivation, and Differentiation of Adipose Tissue-Derived Mesenchymal Stromal/Stem Cells (ASCs) 

ASCs used in this study were isolated from human lipoaspirate. Tissue donation was approved by the ethics committee of the Rostock University Medical Center (http://www.ethik.med.uni-rostock.de/) under the registration number A2013-0112. It complies with the ethical standards of the World Medical Association Declaration of Helsinki. Informed consent was obtained from all patients. The material for the different analyses was retrieved from four patients. Data, as supplied by the surgeons, showed one male and three female patients. Patients were on average 39.5 years old (ranging from 32 to 47). Liposuction procedures were performed by waterjet-assisted liposuction or a tumescent suction technique.

ASC isolation was performed as previously described [12]. ASCs were cryopreserved in passage 2 until they were used for the experiments. To achieve this, the cell suspension was transferred into a cryovial (Greiner bio-one, Germany) containing 150 µL DMSO (Sigma-Aldrich, Germany) and 350 µL fetal calf serum (FCS, PAN Biotech, Germany). The cryovials were cooled to −80 °C overnight at 1 °C per minute in a freezing container (Thermo Fisher Scientific, Berlin, Germany). Vials were then stored in liquid nitrogen at −165 °C until further use. For use in the planned experiments, the cells were thawed stepwise by gently shaking the vial at 37 °C in a water bath for 1 min. The cell suspension was then gradually transferred into cell culture medium (Dulbecco’s Modified Eagle Medium/DMEM GlutaMAX-I, Thermo Fisher Scientific, Germany), containing 10% FCS and antibiotics (100 U/mL penicillin, 100 mg/mL streptomycin, Thermo Fisher Scientific, Germany; hereinafter called maintenance medium) at room temperature and centrifuged for 5 min at 400× *g*. The resulting cell pellet was resuspended in maintenance medium and centrifuged again for 5 min at 400× *g*. Cells were resuspended in maintenance medium and seeded onto cell culture flasks for cultivation at 37 °C and 5% CO_2_ in a humidified atmosphere. When confluency was reached after 5 d, cells were detached from the cell culture flasks by incubation with 0.25% Trypsin EDTA for 5 min at 37 °C. The cells were seeded into cell culture petri dishes (Greiner bio-one, Frickenhausen, Germany) at a density of 20,000 cells/cm^2^. After 24 h of incubation the medium was replaced. After further 48 h (Day 0), additional to the nondifferentiating cultures, adipogenic differentiation of ASCs was induced by adding a differentiation-stimulating medium: i.e., a maintenance medium containing 1 μM dexamethasone, 500 μM IBMX, 200 μM indomethacin, and 10 μM insulin (Sigma-Aldrich, Munich, Germany). Adipogenic stimulation (AS) took place with every replacement of medium three times a week (every second or third day). The medium replacement of differentiating ASCs and undifferentiating ASCs without specific differentiation factors was performed (unstimulated/US) simultaneously. To also assess the emissions of the cell culture medium, pure culture medium without ASC (medium control) was treated and analyzed in a manner identical to the ASC cultures.

### 2.2. Analysis of Cell Numbers

Cell numbers were determined according to the manufacturer’s instructions using the Nucleocounter NC200 and the Via1-Cassettes™ (ChemoMetec, Lillerod, Denmark).

### 2.3. Analysis of Adipogenic Differentiation

In order to assess the adipogenic differentiation of the ASCs over the course of the experiment, fluorescent staining of the nuclei and lipid-filled vacuoles was done at the corresponding time points of 1, 7, 14, and 21 d. Cells were seeded into 96-well µClear^®^ cell culture plates (Greiner, Frickenhausen, Germany) at a density of 20,000 cells/cm^2^. Media changes were done according to the same schedule as the VOC quantification. At the corresponding time points the cells were washed twice with PBS and fixed with 4% paraformaldehyde for 30 min at room temperature. Thereafter, the cells were incubated with a Bodipy/Hoechst-staining solution (100 µL of 1 µg/mL Bodipy (Life Technologies, California, Carlsbad, USA) and 5 µg/mL Hoechst 33,342 (AppliChem, Darmstadt, Germany) in 150 mM NaCl) for 10 min at room temperature in the dark. After incubation, the cells were washed twice with PBS and twice with water for injection. Subsequently, pictures were taken with the microscopic plate scanning system Hermes WiScan (IDEA Bio-Medical, Rehovot, Israel) with a 10-fold magnification objective.

### 2.4. Volatile Organic Compound (VOC) Sampling by Means of Needle Trap Micro Extraction

For headspace sampling, three cell culture dishes (without lid) containing culture medium and ASCs were introduced into a hermetically closed sampling box under a sterile hood (schematically shown in Figure 1). The sampling box was constructed from emission-free materials (Teflon^®^ and glass) to ensure a reliable measurement of trace VOC profiles [28]. As a negative control, three cell culture dishes containing pure cell culture medium without ASCs were analyzed in parallel in a second box. After introduction of the cell culture dishes, the boxes were flushed with 3 L of clean synthetic air (containing 75% N_2_, 20% O_2_, and 5% CO_2_; Air Liquide, Düsseldorf, Germany). After an incubation time of 60 min at 37 °C in an incubator, headspace sampling by means of NTME was done. A detailed description of the sampling setup was described before [23]. Headspace samples were taken at day 4 after seeding the cells, which means 24 h after changing from maintenance to adipogenic differentiation medium and on day 7, 14, and 21 of differentiating cultures, nondifferentiating cultures, and medium control dishes, respectively (see experimental setup in Figure 1). 

NTME was done with needle trap devices (NTDs, needleEX), obtained from Shinwa Ltd. (Kyoto, Japan). The NTDs were equipped with 3 cm of a copolymer of methacrylic acid and ethylene glycol dimethacrylate. Before sampling, NTDs were preconditioned for 30 min at 200 °C in a heating device (PAS Technology Deutschland GmbH, Magdala, Germany) under a helium flow. Teflon caps (PAS Technology Deutschland GmbH, Magdala, Germany) were used to seal the NTDs before and immediately after collecting the headspace samples. 

For VOC preconcentration, NTDs were connected to a 1 mL sterile singleuse syringe (Omnifix-F, B. Braun Melsungen AG, Melsungen, Germany) and pierced through the septum of a Luer Lock cap (IN-Stopper, B. Braun Melsungen AG, Melsungen, Germany) at one connection port of the sampling box. Bidirectional headspace sampling was done by filling and releasing 1 mL of headspace gas 20 times, as described previously [20]. 

### 2.5. Gas Chromatography and Mass Spectrometry (GC-MS) Analysis

For VOC analysis, an Agilent 7890 A gas chromatograph coupled to an Agilent 5975 C inert XL MSD (Agilent, Santa Clara, CA, USA) with a triple axis detector was used. At an injector temperature of 200 °C, VOCs were thermally desorbed from the NTDs. Sample injection was operated in splitless mode (60 s splitless). The GC was equipped with a 60 m RTX-624 column (0.32 mm ID, 1.8 μm column thickness). Helium carrier gas flow was constant at 1.5 mL min^−1^. The temperature program was as follows: 40 °C for 5 min, 8 °C min^−1^ to 120 °C for 2 min, 10 °C min^−1^ to 220 °C, and 20 °C min^−1^ to 240 °C for 4.5 min. Analysis of the samples was performed via electron impact ionization (EI—70 eV) in full scan mode, a mass range of 35–250 amu, and a scan rate of 2.73 scan/s. Volatile organic compounds were tentatively identified by mass spectral library search (NIST Version 2.0). Substance attribution was confirmed by comparing GC retention times and mass spectra of all selected substances with those of pure reference substances. Quantifications were made in parts per billion per volume (ppbV) and then calculated to nmol/L. To quantify the detected substances, a six-point calibration curve (from 1 to 500 ppbV) was established. Humidity-adapted standards were prepared by means of a liquid calibration unit (LCU, Ionicon Analytik GmbH, Austria). Limits of detection were calculated as a signal-to-noise ratio of 3:1, whereby limits of quantification corresponded to a signal-to-noise ratio of 10:1. Ten blank NTDs were analyzed for this purpose. Quantitative parameters (limit of detection (LOD), limit of quantification (LOQ), and standard deviation (SD)) for all identified VOCs are shown in Appendix A.

### 2.6. Reference Substances

Acetone, 2-butanone, heptanal, octanal, and acetaldehyde were acquired from Ionimed Analytik GmbH (Insbruck, Austria). Benzaldehyde and 1,3-di-tert-butylbenzene were purchased from Sigma-Aldrich (Germany); tert-butanol, ethylbenzene, as well as 2-ethylhexanol were purchased from TCI (Eschborn, Germany) and pentane from Fluka (Munich, Germany).

### 2.7. Statistical Analysis

Figure 2, Figure 3, Figure 4 and Figure 5 show numerical data represented as medians and percentiles (25–75%). Correlation analysis and statistical testing was performed by using RStudio (version 1.0.136) and R software (version 3.3.2_ 2016-10-31). By using the Kruskal–Wallis test with the Nemenyi post hoc test, statistically significant differences in median values between all groups were identified. Values of *p* < 0.05 were considered statistically significant.

## 3. Results

### 3.1. Adipogenic Differentiation of ASCs

The assessment of adipogenic differentiation by fluorescent lipid staining revealed that nondifferentiating conditions did not induce lipid accumulation in ASCs at any time point (Figure 2a,c,e,f), After 24 h, adipogenic stimulation did not lead to lipid accumulation (Figure 2b). After 7 d of adipogenic stimulation, distinct lipid vacuole formation was visible (Figure 2d). With the progression of adipogenic differentiation of ASCs (day 14 and 21), the lipid vacuoles appeared larger and showed a higher fluorescence intensity (Figure 2f,h). Thus, considerable adipogenic differentiation was clearly visible in almost all cells of the adipogenic cell culture model. This was not the case in nondifferentiating ASCs.

### 3.2. Comparision of VOC Emissions of Non-Differentiating and Adipogenically Differentiating ASCs

Thirty-one potential volatile marker substances were identified in the headspace above the cell cultures (concentration ranges and quantitative parameters are depicted in Appendix A). To focus on the most relevant substances, we excluded substances having concentrations below the limit of detection (LOD) at more than 3 time points as well as substances having concentrations in the same range (± 10%) as in pure media samples. Based on these criteria, 11 VOCs were selected for further analysis. These substances were identified and quantified by using pure reference substances. Six of these VOCs that were differentially emitted and dependent on the differentiation are depicted in Figure 3. 

The compounds acetaldehyde, pentane, and 1,3-di-tert-butylbenzene displayed different trends of emission during cultivation with maintenance media compared with cultivation with differentiation medium (Figure 3). 

Concentrations of the emitted acetaldehyde changed significantly between days 7, 14, and 21 of adipogenically differentiated ASCs. In nondifferentiating ASCs, no significant concentration differences could be detected between the days of cultivation.

Pentane concentrations decreased in nondifferentiating ASCs, whereas a slight increase in adipogenically differentiating ASCs was detected from day 14 to day 21 of differentiation. The production of pentane in the adipogenically differentiating cell cultures peaked on day 1 and was significantly higher than on day 7 and on day 14. 

The concentration of 1,3-di-tert-butylbenzene in the headspace of nondifferentiating ASCs remained lower as well as constant over the time, whereas the emissions in the headspace of adipogenically differentiating cell cultures showed a slight increase.

Concentrations of ethylbenzene and benzaldehyde differed between nondifferentiating and adipogenically differentiating ASCs in terms of concentration and time course. Both substances showed lower emissions in adipogenically differentiating ASCs compared with nondifferentiating ASCs. For benzaldehyde, these differences were significant. Heptanal showed a significant decrease over time in nondifferentiating ACSs and an increase in adipogenically differentiating ASCs. Concentrations of heptanal on day 21 of cultivation were higher in adipogenically differentiating compared with nondifferentiating ASCs. Detailed statistical information can be found in the Appendix A. 

To emphasize the differences between nondifferentiating and adipogenically differentiating ASCs, Table 1 presents concentrations of emitted VOCs normalized to the cell number at day 21. All substances other than benzaldehyde showed higher concentrations in differentiating cells than in undifferentiating ASCs. The most prominent differences were found for acetaldehyde and heptanal. In both substances, a nearly threefold increase of emitted concentrations could be detected on day 21 of the experiment.

### 3.3. Comparision of VOC Emissions from Medium Control and Corresponding Cell Cultures 

Apart from emissions of the cells, VOCs were also emitted from the cell culture medium without cells. Therefore, the headspace of the medium without cells was used as a “medium control” for each time point tested. The detailed depictions of emissions of differentiating and nondifferentiating ASCs compared to the medium control are shown in Appendix A, respectively. In the following, exemplary results from adipogenically differentiating ASCs are presented.

Acetaldehyde, pentane, and 1,3-di-tert-butylbenzene showed medium independent emissions in adipogenically differentiating cultures. Acetaldehyde and 1,3-di-tert-butylbenzene emissions increased at every measurement time point. Most other VOCs (especially aldehydes) showed lower and decreasing concentrations compared to the medium controls. 

#### 3.3.1. VOC Consumption during Adipogenic Differentiation 

The emissions of three aldehydes (heptanal, octanal, and benzaldehyde) and one aromatic hydrocarbon (ethylbenzene) were higher in the pure cell culture media compared with cell cultures. As examples, ethylbenzene and benzaldehyde of this VOC group are depicted in Figure 4 (heptanal and octanal can be found in Appendix A). 

Ethylbenzene concentrations were significantly higher in medium controls compared with cell culture samples at three time points analyzed. For benzaldehyde, we could find significant differences in the emissions from all cell culture samples compared with the pure medium control at all time points. In nondifferentiating ASCs, ethylbenzene and benzaldehyde emissions showed the same profile of higher emissions in the medium control compared with the cell culture (see Appendix A). Thus, the specific consumption or binding of these VOCs by the cells is indicated.

#### 3.3.2. Culture Medium-Dependent VOCs

The emissions of 2-ethylhexanol, acetone, tert-butanol, and 2-butanone from adipogenically differentiating ASCs complied with the emissions of the medium controls. Heptanal and octanal emissions also appeared media-dependent in nondifferentiating ASCs (see Appendix A).

The emissions of two exemplary VOCs, acetone and 2-ethylhexanol, over 21 days of cultivation in adipogenically differentiating ASCs are presented in Figure 5. The emissions of tert-butanol and 2-butanone for adipogenically differentiating ASCs showed a slight decreasing trend over the first two weeks of differentiation and peaked at day 21. This trend was similar in the medium control (Appendix A). The more comprehensive depiction of adipogenically differentiating and nondifferentiating ASC VOCs from Figure 4 and Figure 5 can also be found in the Appendix A.

## 4. Discussion

A nondestructive technique that could save time and material and gain more detailed information on cellular metabolism would be desirable. The analysis of VOCs by means of NTME preconcentration of headspace air coupled with GC-MS could be a fast and nondestructive method for the analysis of metabolic processes in cell cultures [29,30,31].

In this study, our previously developed versatile sampling system [28] was applied to monitor VOC emissions of cellular differentiation under standardized conditions. This method, however, presents some challenges. Physical, chemical, and biological processes (e.g., the degradation of FCS in culture medium) might affect VOC emissions. Therefore, the analysis of an adequate control—in this case pure cell culture medium without cells—is a crucial step in order to obtain reliable results. Large variations in VOC concentrations emitted from pure culture medium have been observed previously [21] and might be due to physical and chemical processes during heating and cooling as well as aging of the media. Furthermore, lipophilic VOCs (such as ethylbenzene, heptanal, or 2-ethylhexanol) might also be bound to lipids of cellular structures (e.g., membranes) in general [32].

In our study we utilized an established cell culture model that showed clear differences between nondifferentiated cells and cells in adipogenic differentiation. Analysis of the formation of lipid-filled vacuoles clearly confirmed the differentiation of the ASCs after adipogenic stimulation, whereas the nondifferentiated ASCs did not show any signs of adipogenesis. This cell culture model has already been examined for energy metabolic aspects, and it has been shown that adipogenic differentiation induces a shift towards lipid metabolism [12].

In the headspace of both ASC cultures (adipogenically differentiating and nondifferentiating), some VOC emissions seem to be related to culture media emissions rather than to emissions from the cells in culture. This applies to four substances (2-ethylhexanol, acetone, tert-butanol, and 2-butanone) in the headspace of adipogenically differentiating ASC cultures and two substances (heptanal and octanal) in the headspace of nondifferentiating ASC cultures. Whether these changes are specific and biochemically driven or merely induced by unspecific chemical conversion remains unclear. However, since the VOCs acetaldehyde, pentane, and 1,3-di-tert-butylbenzene showed clear emission differences dependent on the differentiation conditions, a specific link to metabolic processes might be possible.

Our data indicated that ASCs undergoing adipogenic differentiation increased the emission of acetaldehyde, whereas nondifferentiated ASCs reduced the emissions down to almost zero. Acetaldehyde emissions were nearly three times higher on day 21 of cultivation in adipogenically differentiating ASCs compared with nondifferentiating cultures. When taking into account the cell number, which is lower in adipogenically differentiating ASCs due to a stop in proliferation [12], this effect becomes even more distinctive. Thus, in our study, acetaldehyde is the VOC with the most variable differentiation-dependent behavior. 

Acetaldehyde might be connected to mitochondrial pyruvate decarboxylation [33,34]. Whether acetaldehyde is consumed or produced could indicate the demands for intracellular acetyl-CoA. The energy metabolism of nonstimulated cells and adipogenically stimulated ASCs might be based predominantly on carbohydrate oxidation with a flux of pyruvate to acetyl-CoA. Under these conditions, it is possible that acetaldehyde can be used as well for acetyl-CoA production.

However, acetaldehyde can also be the product of ethanol degradation. Since some differentiation-inducing additives are dissolved in ethanol, a general, slightly increased level of acetaldehyde emissions compared with ASCs in maintenance medium might result from higher ethanol levels in the differentiation medium. This might be reflected in the increase of acetaldehyde emissions in the medium controls at an early time point (up to seven days). Ethanol dehydrogenase activity was detected in brown adipose tissue of rats [35]. Whether ethanol dehydrogenase is also present in the ASCs of white adipose tissue needs to be determined.

1,3-di-tert-butylbenzene emissions showed a significant increase in adipogenically differentiating ASCs compared to nondifferentiating ASCs. Tang et al. [36] found 1,3-di-tert-butylbenzene emitted differentially in cancer cells lines compared with nonmalignant cells. No further information is available about this substance in the current literature.

When normalized to cell numbers, pentane concentrations were higher in adipogenically differentiating ASCs than in nondifferentiating ones. Pentane is produced through the peroxidation of lipids and other biomolecules under the action of reactive oxygen species [37]. As shown before, adipogenic differentiation of ASCs correlates with an increase of the antioxidative defense activities of glutathione reductase [12]. This could conform with our current findings, which indicate an increase in the production of reactive oxygen species in adipogenically differentiating cells, and which could be followed by an increase in the antioxidative defense systems of ASCs.

Ethylbenzene showed higher concentrations in the headspace of medium controls than in the headspace of cultivated cells. In general, lower and less-deviating concentrations were observed in the headspaces of adipogenically differentiating ASCs compared with nondifferentiating ASCs. We, thus, conclude that cells in culture with and without differentiation stimulation consume or bind this compound. Previous studies established that ethylbenzene undergoes α- and ω-oxidation [38].

The aldehydes benzaldehyde, heptanal, and octanal were also consumed or bound by the cells from the culture media under both cultivation conditions. A consumption of aldehydes from the culture media might be based on the activity of aldehyde dehydrogenase [39,40] (e.g., benzaldehyde concentrations were reduced in the headspace of all cells but were even lower in adipogenically differentiating than in nondifferentiating cells). Zimmermann et al. [41] described NAD-aldehyde dehydrogenase as a possible enzyme for metabolizing benzaldehyde in fatty acid metabolism and tryptophan metabolism [42,43]. Amino acids are necessary for proliferation and differentiation, and their consumption was also observed in stem cell lines [27]. 

Heptanal and octanal showed relatively higher concentrations on day 21 in the adipogenically stimulated ASC cultures when related to the cell number. Since heptanal can be a product of lipid peroxidation [44,45], increasing enzyme activity for lipid oxidation during differentiation could lead to a higher turnover of heptanal. Other studies linked octanal to oxidative stress [46,47]. While this process includes lipid oxidation, octanal might increase during adipogenic differentiation due to an increase of the oxidation processes.

## 5. Conclusions

In this study we were able to identify differentially emitted VOCs, which might be linked to a changed metabolic activity in ASCs dependent on cellular differentiation. Whether these changes are driven by specific, enzyme-mediated processes or are the result of unspecific reactions has to be investigated. Irrespective of the fact that these are early stage results, the data presented are in agreement with the results of adipogenic-differentiating cells to date. Thus, further experiments might reveal if headspace VOC measurements serve as a marker for the quality and quantity of adipogenic differentiation of ASCs.

## Figures and Tables

**Figure 1 cells-08-00697-f001:**
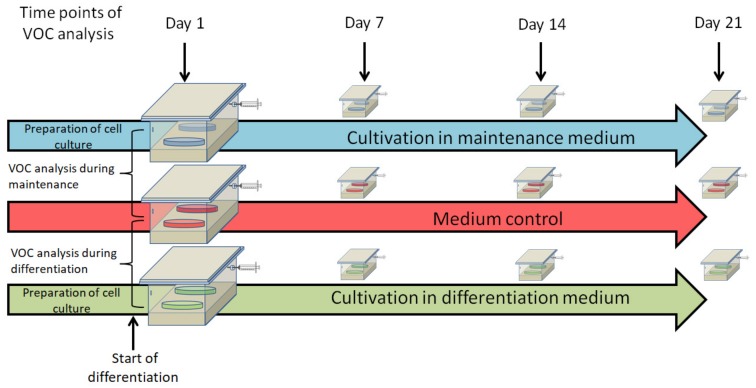
Schematic depiction of the experimental setup. VOC—volatile organic compound.

**Figure 2 cells-08-00697-f002:**
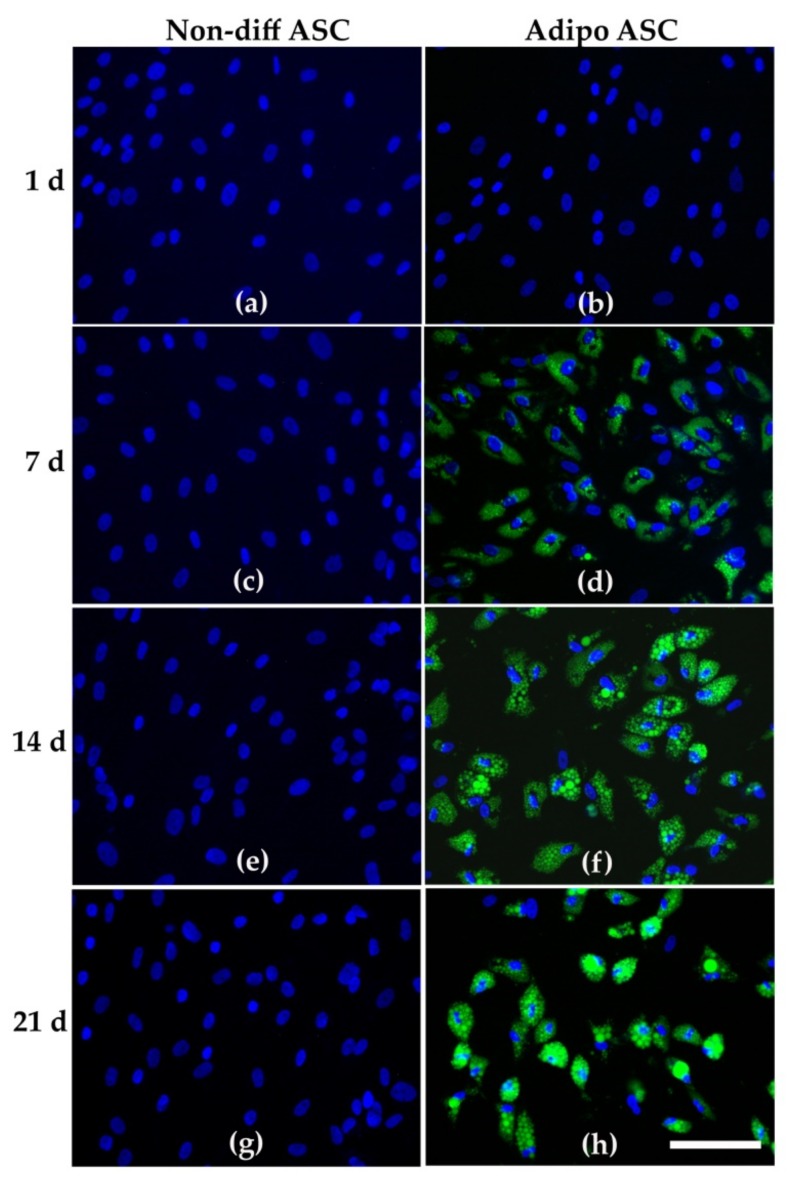
Depiction of adipose tissue-derived mesenchymal stromal/stem cells (ASCs) under nondifferentiating conditions (referred to as non-diff ASC, subfigures (**a**,**c**,**e**,**g**) for cultivation days 1, 7, 14, 21, respectively) and under adipogenic stimulation (adipo ASC, subfigures (**b**,**d**,**f**,**h**) for cultivation days 1, 7, 14, 21, respectively) Ffluorescence staining of nuclei (blue) and lipids (green), scale bar: 100 µm.

**Figure 3 cells-08-00697-f003:**
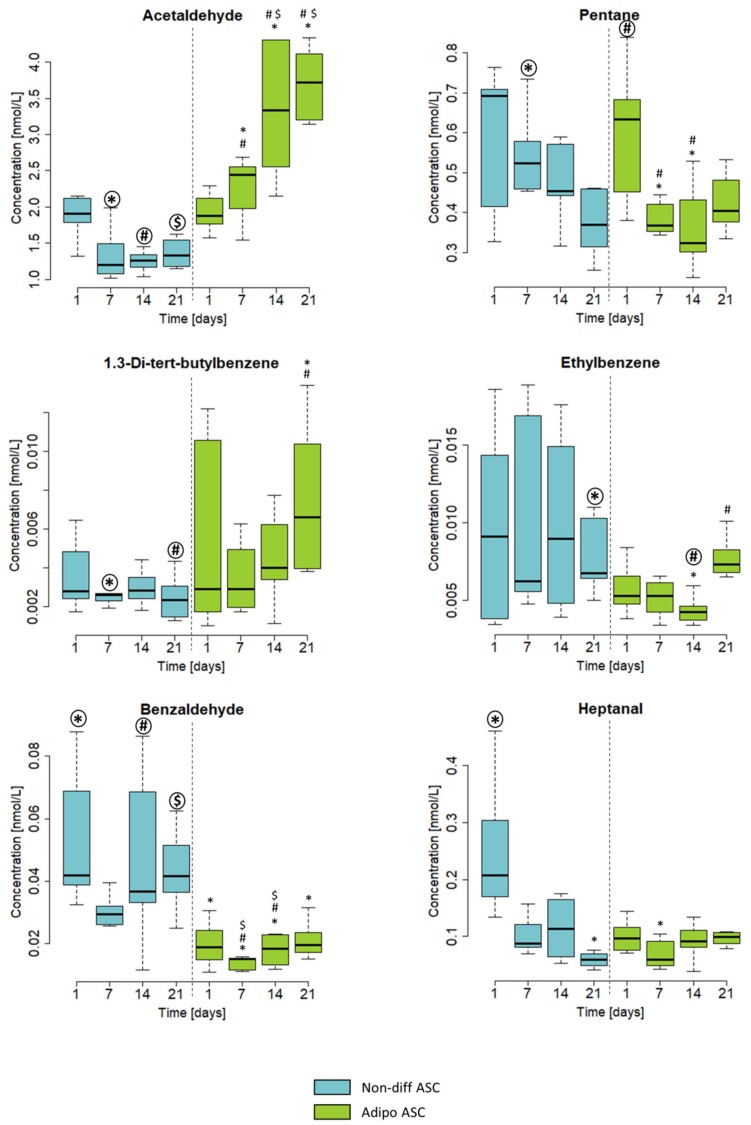
Emissions of acetaldehyde, pentane, 1,3-di-tert-butylbenzene, ethylbenzene, benzaldehyde, and heptanal from nondifferentiating (non-diff ASC, blue) and adipogenically differentiating ASC (adipo ASC, green). Concentrations in the headspace are shown in nmol/L on the Y-axis. The X-axis shows the time points of measurements. The boxplots represent data from three independent experiments. Significance was tested within all groups. Symbols (*, #, $) indicate significant differences to the corresponding highlighted group (*p*-values < 0.05).

**Figure 4 cells-08-00697-f004:**
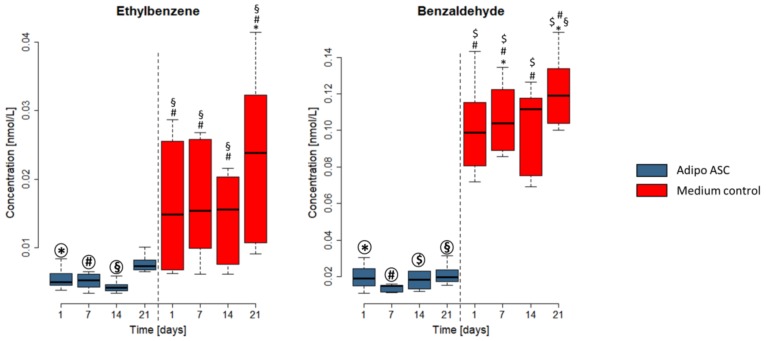
Example of two VOCs showing lower emissions in adipogenic differentiating ASC (adipo ASC) cultures than in the culture media controls. Concentrations in the headspace in nmol/L are shown on the Y-axis. The X-axis shows the time point of measurements. The diagram shows cell culture with adipogenically differentiating ASC (referred to as Adipo ASC, in blue) and medium controls (samples without cells, in red). Boxplots represent the data from three independent differentiation experiments. Significance was tested within all groups. Symbols (*, #, §, $) indicate significant differences to the corresponding highlighted group (*p*-values < 0.05).

**Figure 5 cells-08-00697-f005:**
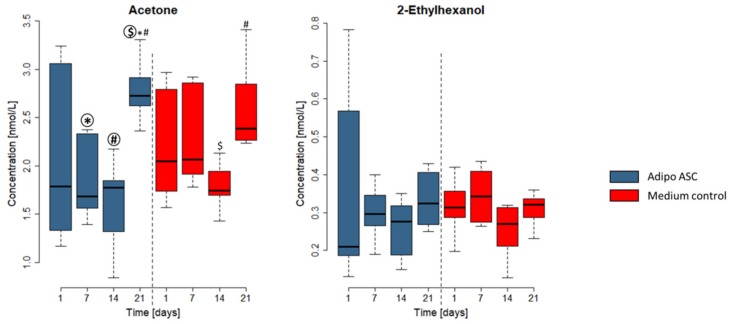
Acetone and 2-ethylhexanol as exemplary medium-dependent VOCs in adipogenically differentiated ASCs (referred to as Adipo ASC). The concentration in the headspace in nmol/L is shown on the Y-axis. The X-axis shows the measurement time points. The diagram shows cell culture with adipogenically differentiating ASC (referred to as Adipo ASC, in blue) and medium controls (samples without cells, in red). Boxplots represent data of three independent differentiation experiments. Significance was tested within all groups. Symbols (*, #, $) indicate significant differences to the corresponding bold highlighted group (*p*-values < 0.05).

**Table 1 cells-08-00697-t001:** Ratio of VOC production in the headspace per cell from nondifferentiating and adipogenically differentiating ASCs at day 21 of cultivation. Concentrations were calculated per 1 × 10^6^ cells from one experiment as an example. VOC concentrations are shown in pmol per L (pmol/L) ± standard deviation (SD). Statistical significance was tested by a *t*-test (*p* < 0.05).

Substance	Normalized VOC Concentration [pmol/L per 1 × 10^6^ cells ± SD]		Statistically Significant
	Nondifferentiating ASCs	Adipogenically Differentiating ASCs	
Acetaldehyde	229.3 (± 34.5)	686.4 (± 14.4)	Yes
Pentane	64.2 (± 8.7)	86.9 (± 1.1)	Yes
1,3-bis-(1,1-dimethylethyl) Benzene	0.368 (± 0.04)	1.4 (± 0.2)	Yes
Ethylbenzene	0.94 (± 0.1)	1.9 (± 0.4)	Yes
Benzaldehyde	6.17 (± 2.1)	5.93 (± 1.2)	No
Heptanal	9.6 (± 1.9)	25.4 (± 5.1)	Yes
Octanal	11.5 (± 1.1)	23.8 (± 0.6)	Yes

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
