# Peer review of "Differences in the Emission of Volatile Organic Compounds (VOCs) between Non-Differentiating and Adipogenically Differentiating Mesenchymal Stromal/Stem Cells from Human Adipose Tissue"

_cells, 2019, doi:10.3390/cells8070697_

Round 1

Reviewer 1 Report

The authors study volatile organic compound (VOC) emissions in the headspace of cell cultures in order to acquire information on the metabolic changes of adipose tissue-derived mesenchymal stromal/stem cells (ASC) during their adipogenic differentiation. They seek to answer to the question of whether they can detect differences in VOC emission from cell cultures between non-differentiating and differentiating ASC, and consequently, whether these differences in the emission of VOC’s [acetaldehyde, pentane, 1.3 bis (1.1 dimethylethyl) benzene, benzaldehyde, heptanal, octanal, shown in Table 1] can be related to metabolic changes in ASC during differentiation. Their long turn aim is to establish the use of headspace VOC measurements as a marker for the quality and quantity of adipogenic differentiation of ASC. The authors have an extensive experience in this area of research, their experiments are well documented with appropriate controls, the manuscript is concisely written and supported by appropriate references, the results are clearly presented in figures and tables, extended to the Supplement), and the conclusions drawn are valid.

Author Response

Reviewer 1

Thank you, for your comments and your positive feedback!

Please find attached the manuscript with all changes suggested by all reviewers.

Reviewer 2 Report

The present manuscript presents interesting data besides dealing with an important issue: the development on an non-invasive method, based on VOC analysis,  to characterise and discriminate non-differentiating from differerentiated adipogenically MSCs from adipose tissue. However, the English should be reviewed. For example, the information at the introduction section is not so well integrated. The structure pointed at the results section is not intuitive. Now the structure, is:

3.1 adipogenic differentiations

3.2 VOC emissions from cell culture

3.3 VOC emissions from medium culture without cells

3.3.1 VOC consumed during differentiation

3.3.2 Culture medium dependent VOC

Would simplify data understanding, if VOC concentrations observed in the medium without cells (that should consider as control), would be represented in the same graph of VOCs present at the cells culture with and without differentiation.

Some other minor comments:

-          The mode of referentiation should be reviewed. For ex. Instead of “[14]–[16].” should be [14-16], and instead of “[10], [11].” Should be “[10, 11].”

-          In Introduction section, line 41, authors refer that “In recent years…”, however, the 3 references already have more than 17, 15 and 10 years, respectively.

-          Information of paragraph starting on line 68, should be more detailed. For example, should be indicated that this is from stem cells, and should be contextualised why referring to ethanol and acetaldehyde.

-          In legend of Fig3, instead of VOC the substances should be referred as conducted in legend of fig. 4.

-          In legend of Fig. 3 and 4, it is not understandable which group of data is being statistically compared, i.e. to what group of experiments the *,# and $ are referring to?

-          Due to its relevance, the heptanal concentrations should also be included in a figure as fig 3 and 4 for other compounds.

-          Legend of Table 1, should indicate that refers to 21 days of culture. This information should be consequently deleted from the table itself.

-          Table 1, should indicate if the differences between non differentiated and differentiated were statistically different.

Reviewer 3 Report

In this work, a non-invasive analysis of human adipose tissue-derived mesenchymal stromal/stem cell differentiation through metabolic monitoring of its volatile organic compounds (VOCs) has been reported. Overall, the manuscript is well-explained. However, some major and minor issues should be clearly addressed prior to the publication, as written below:

(Major issues)

1. Introduction is too short, the authors could add more references regarding on the lacks of common methods, due to support the superiority of this work

2. Needle trap micro-extraction is not new anymore in terms of biological applications. Thus, please clearly point out the novelty of this proposed technique

3. Since GC-MS was complementary used in this method, thus the GC-MS spectra are better presented in the main figures as one aligned graph for initial data.

(Minor issues)

1.        In the case of Figure 3 and 4, if possible, both of them can be simplified into one figure, since they seem to be same typical data

2.        Please pay attention to state a numerical statement, it should be following on scientific form

3.        Some errors were found as follows: page 6 line 196 (figure e and f --> figure e and g); page 9 line 267 (consumed --> consumption)

4.        Correct all the punctuations and grammatical error in this manuscript

Round 2

Reviewer 2 Report

The present manuscript presents interesting data besides dealing with an important issue: the development on an non-invasive method, based on VOC analysis,  to characterize and discriminate non-differentiating from differerentiated adipogenically MSCs from adipose tissue

Reviewer 3 Report

The revised MS is now acceptable for publication in the journal.